# Revisiting foundation models for cell instance segmentation

**Anwai Archit**[1] (iD)                                        ANWAI.ARCHIT@UNI-GOETTINGEN.DE

**Constantin Pape**[1,2,3] (iD)              CONSTANTIN.PAPE@INFORMATIK.UNI-GOETTINGEN.DE

[1] *Georg-August-University Göttingen, Institute of Computer Science*

[2] *CAIMed - Lower Saxony Center for AI & Causal Methods in Medicine, Göttingen*

[3] *Cluster of Excellence Multiscale Bioimaging (MBExC), Georg-August-University Göttingen*

**Editors:** Accepted for publication at MIDL 2026

## Abstract

Cell segmentation is a fundamental task in microscopy image analysis. Several foundation models for cell segmentation have been introduced, virtually all of them are extensions of Segment Anything Model (SAM), improving it for microscopy data. Recently, SAM2 and SAM3 have been published, further improving and extending the capabilities of general-purpose segmentation foundation models. Here, we comprehensively evaluate foundation models for cell segmentation (CellPoseSAM, CellSAM, $\mu$SAM) and for general-purpose segmentation (SAM, SAM2, SAM3) on a diverse set of (light) microscopy datasets, for tasks including cell, nucleus and organoid segmentation. Furthermore, we introduce a new instance segmentation strategy called automatic prompt generation (APG) that can be used to further improve SAM-based microscopy foundation models. APG consistently improves segmentation results for $\mu$SAM, which is used as the base model, and is competitive with the state-of-the-art model CellPoseSAM. Moreover, our work provides important lessons for adaptation strategies of SAM-style models to microscopy and provides a strategy for creating even more powerful microscopy foundation models.

**Keywords:** vision foundation models, segment anything, microscopy, instance segmentation, cell segmentation

## 1. Introduction

Instance segmentation is one of the most important image analysis tasks in microscopy, enabling phenotypic drug screening in high-content imaging (Chandrasekaran et al., 2024), analysis of embryogenesis at the cellular level (Lange et al., 2024), and many other applications. Virtually all current methods for microscopy instance segmentation are based on deep learning, such as dedicated tools addressing cell (Stringer et al., 2021) and nucleus (Schmidt et al., 2018) segmentation in light microscopy, nucleus segmentation in histopathology (Graham et al., 2019), or segmentation of mitochondria (Conrad and Narayan, 2023) and other organelles (Muth et al., 2025) in electron microscopy. A repository (Ouyang et al., 2022) collects trained models for such segmentation tasks, compatible with tools for model inference (Gómez-de Mariscal et al., 2021; Berg et al., 2019).

These dedicated models can accelerate many analyses, yet the large number of tools and the potential lack of pretrained models for specific tasks put a burden on users without substantial computational expertise. Hence, foundation models have been introduced in this domain, enabling a wider range of tasks with a single model (Archit et al., 2025a;

Pachitariu et al., 2025; Marks et al., 2025; Hörst et al., 2024; Griebel et al., 2025). They are predominantly based on the Segment Anything Model (SAM) (Kirillov et al., 2023), a general-purpose segmentation foundation model. SAM itself has been extended to video data by SAM2 (Ravi et al., 2025) and, recently, to text- and example-based segmentation by SAM3 (Carion et al., 2025). The latter also included microscopy data in its training set.

These developments open up the following questions: (i) What is the best strategy for adapting a SAM-style model to microscopy? (ii) Are specific (foundation) models for microscopy segmentation still needed or do general-purpose segmentation models, in particular SAM3, make them obsolete? (iii) What influence does the training data (modalities, size, data diversity) have on model performance?

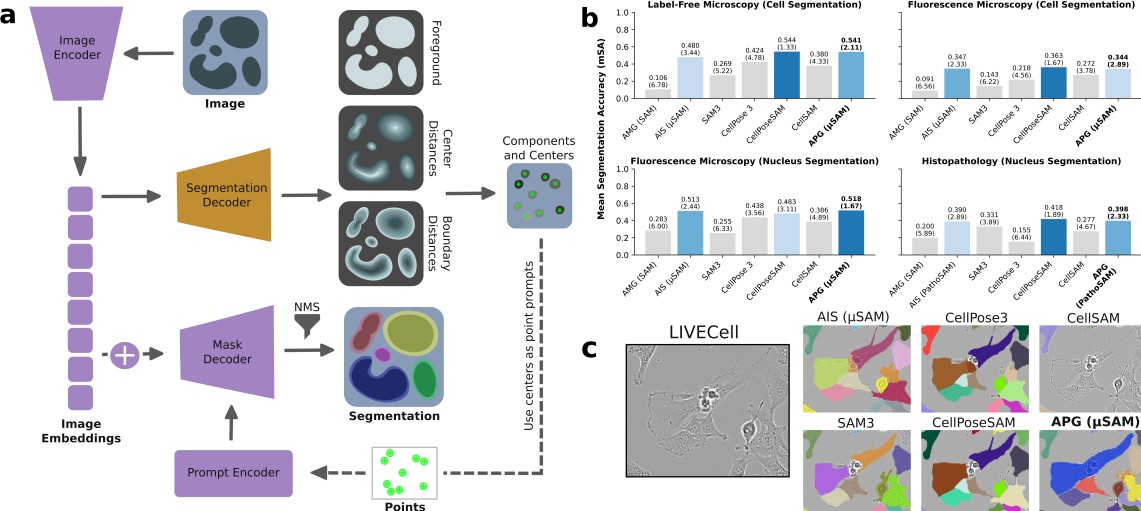

Figure 1: a) Overview of our new instance segmentation method, automatic prompt generation (APG), which re-purposes the trained $\mu$SAM (or PathoSAM) models by deriving point prompts from decoder predictions, predicting masks based on these prompts, and then filtering overlapping masks via NMS. Note that the model is not retrained. APG replaces the prior instance segmentation logic. b) Overview of segmentation results for four different microscopy modalities. We report the averaged rank over the 9 datasets per domain in parentheses, top three methods are colored. c) Example label-free cell segmentation with different methods. Only APG correctly segments the large central cell, highlighting its advantage for complex cell morphologies.

To address these questions we: (i) Extensively benchmark recent foundation models for general-purpose and microscopy segmentation. (ii) Develop a new instance segmentation algorithm that operates on top of (fine-tuned) SAMs without the need for additional training. Our results show substantial improvements thanks to our segmentation approach, competitive with the state-of-the-art. Further, they show that foundation models for microscopy

have an edge over SAM3, and show a substantial influence of the training data on model performance. See Fig. 1 for an overview of our methodology and contributions.

## 2. Related Work

Vision foundation models (VFMs) have emerged after the success of large language models (LLMs) as generally capable language processors (Brown et al., 2020). VFMs can be divided into three categories: (i) vision and text models, such as CLIP (Radford et al., 2021) and SigLIP (Zhai et al., 2023) that are trained via contrastive learning on image-text-pairs. They are a key ingredient of multi-modal LLMs. (ii) general-purpose vision encoders such as Dino V2 (Oquab et al., 2024) and V3 (Siméoni et al., 2025) that are trained via self supervised learning, enabling diverse downstream tasks by fine-tuning small decoders. And (iii) foundation models for segmentation. Among these, SAM (Kirillov et al., 2023) is the most popular. It supports interactive segmentation based on point or box prompts, and also supports automatic segmentation. SAM2 (Ravi et al., 2025) extends its capabilities to video segmentation. SAM3 (Carion et al., 2025) introduces "concept-based" segmentation, enabling prompting with short text descriptions or example images with object annotations. Other examples of segmentation foundation models include SegGPT (Wang et al., 2023), which supports example-based learning, LISA (Lai et al., 2024), which extends an LLM with segmentation capabilities, and SEEM (Zou et al., 2023), which offers a wide range of prompting options.

Many segmentation foundation models for biomedical images have emerged. They are predominantly based on SAM(2). Their exact adaptation strategies vary and can be divided into three categories and combinations thereof:

1. By automatically deriving prompts (points or bounding boxes) for objects in the image and then using SAM to predict the corresponding masks (Chen et al., 2024).

2. By using SAM's encoder as backbone of a model for automatic segmentation that is trained on domain-specific data (He et al., 2025).

3. By fine-tuning SAM's architecture for promptable segmentation on domain-specific data (Ma et al., 2024; Archit et al., 2025b; Cheng et al., 2023).

Several segmentation foundation models have been proposed for medical imaging, e.g. CT, MRI, X-Ray, (see citations in the previous paragraph). Instance segmentation in microscopy is another important application. The most popular models are CellSAM (Marks et al., 2025) for light microscopy segmentation, $\mu$SAM (Archit et al., 2025a) for light and electron microscopy segmentation, its extension PathoSAM (Griebel et al., 2025) for histopathology, CellPoseSAM (Pachitariu et al., 2025) for light microscopy and histopathology, and CellViT (Hörst et al., 2024) for histopathology. See also Sec. 3.1.

## 3. Methods

We first review foundation models for microscopy segmentation (Sec. 3.1), then explain our new instance segmentation method (Sec. 3.2), and the evaluation methodology (Sec. 3.3).

### 3.1. Instance segmentation with SAM-based models

The models of the SAM family all support automatic instance segmentation. SAM (Kirillov et al., 2023) and SAM2 (Ravi et al., 2025) are trained with an objective for prompt-based segmentation. They support automatic segmentation by covering the input image with a grid of point prompts, segmenting each prompt, and then merging the predicted masks via non-maximum suppression (NMS). This mode is called automatic mask generation (AMG). In contrast, SAM3 (Carion et al., 2025) predicts instances directly with a DETR-style approach (Carion et al., 2020). All three models were primarily trained on natural images with segmentation annotations, 11 million images with 1 billion annotations for SAM, an additional 50k videos with 642k object track annotations for SAM2, and an additional 5 million images and 50k videos for SAM3.

SAM has been studied extensively for microscopy data (see e.g. (Archit et al., 2025a), Sec. 4). It yields good segmentation results for easy tasks (e.g. well separated nuclei) via AMG, but fails for more difficult tasks. AMG with SAM performs worse for microscopy (see Sec. 4). SAM3 has been published very recently. To our knowledge we are the first to evaluate it for microscopy.

While SAM itself fails at difficult microscopy segmentation tasks, the state-of-the-art models for microscopy are built on top of it, following one of the strategies 1.-3. outlined in Sec. 2. The simplest strategy (2) is to train a new decoder on top of SAM's image encoder (initialized with its pretrained weights) that outputs an intermediate prediction, followed by (non-differentiable) post-processing to obtain instances. This approach is implemented by CellPoseSAM (Pachitariu et al., 2025), which chose the CellPose instance segmentation method (Stringer et al., 2021) and was trained on 22,826 light microscopy and histopathology images with 3.34 million annotated cells and nuclei, and Cell-ViT, which chose the HoverNet semantic instance segmentation method (Graham et al., 2019) and was trained on the PanNuke (Gamper et al., 2020) dataset, consisting of 200,000 annotated nuclei.

CellSAM (Marks et al., 2025) takes a more complex approach: it trains a bounding box detection decoder (CellFinder) on top of SAM's image encoder and then uses its predictions as box prompts for SAM's prompt encoder to segment instance masks. The mask decoder is also finetuned, i.e., corresponding to a combination of strategies 1 and 2. CellSAM was trained on ten datasets with annotated cells and nuclei.

$\mu$SAM (Archit et al., 2025a) finetunes the entire SAM architecture for promptable segmentation while adding a decoder for instance segmentation that predicts foreground probabilities as well as normalized distances to object centers and boundaries. These predictions serve as input to a watershed for instance segmentation. The procedure is called automatic instance segmentation (AIS). $\mu$SAM was trained on ca. 17,000 light microscopy images with over 2 million annotated cells and nuclei; a different version of the model for electron microscopy also exists. PathoSAM (Griebel et al., 2025) replicates this effort for histopathology. It was trained on ca. 5,000 images with over 400,000 annotated nuclei. This approach corresponds to a combination of strategies 2 and 3.

### 3.2. Automatic prompt generation

We observe that none of the SAM-based microscopy foundation models combine all three adaptation strategies, i.e. none of them combine automatically derived prompts (1) with

a custom decoder (2), and finetuning for promptable segmentation (3). While CellSAM comes close to this combination, it does not finetune for promptable segmentation and thus relies heavily on the box predictions, which are translated one-to-one to masks. Hence, if an object is not correctly detected, it cannot be recovered by SAM's promptable segmentation. This becomes more likely under a domain shift, leading to diminished generalization, which could be avoided with a more flexible prompting strategy. On the other hand, segmentation via a dedicated decoder as implemented by CellPoseSAM and $\mu$SAM foregoes potential improvements due to prompt-based segmentation. Empirically, we observed that segmentation based on prompts derived from annotated objects performs significantly better than fully automated segmentation (Archit et al., 2025a). Hence, a suitable automatic prompting strategy should be able to improve upon automatic instance segmentation without prompting, exemplified for a cell with complex morphology in Fig. 1.

To overcome the limitations discussed in the previous paragraph, we implement a new instance segmentation method called automatic prompt generation (APG). It operates on top of the $\mu$SAM model, which was **not retrained** by us. APG uses the predictions of $\mu$SAM's segmentation decoder to derive point prompts, uses the prompt encoder and mask decoder to predict masks based on these prompts, and merges the predicted masks via NMS to obtain an instance segmentation. The procedure is illustrated in Fig. 1 a). Note that this approach does not require us to derive exactly one prompt per object (as in CellSAM) since multiple predicted masks per object can be filtered by NMS. In detail, APG works as follows:

1. Apply the image encoder and segmentation decoder to predict foreground probabilities $fg$ and normalized boundary as well as center distances, $d_b$ and $d_c$.

2. Apply the thresholds $t_{fg}$, $t_b$ and $t_c$ to $fg$, $d_b$, and $d_c$, respectively, to obtain binary masks.

3. Apply connected components to the intersection of the three binary masks from 2.

4. Derive a point prompt for each component by computing the maximum of the boundary distance transform per component.

5. Apply prompt encoder and mask decoder to these prompts to obtain mask and IoU predictions. The latter give a quality estimate for each predicted mask.

6. Apply a size filter $s$ to the masks.

7. Compute the pairwise overlap of predicted masks and apply NMS with threshold $t_{nms}$ based on predicted IoUs to filter overlapping masks.

The parameters of APG are $t_{fg}$, $t_b$, $t_c$, $s$, and $t_{nms}$ . Their default values are: $t_{fg} = 0.5$, $t_b = 0.5$, $t_c = 0.5$, $s = 25$, and $t_{nms} = 0.9$. We run all experiments with these default values.

Note that step 2 and 3 are the same as in the AIS method of $\mu$SAM, which then uses the components as seeds for a watershed. However, in AIS $t_b$ and $t_c$ have to be determined such that each object is covered with a single component, leading to a trade-off between over- and under-segmentation. In APG, we can choose these values so that multiple prompts are derived for one object, then filtered in step 7 by NMS. APG can be applied to the $\mu$SAM

(and PathoSAM) model as is without retraining. APG is implemented as part of the $\mu$SAM code base at https://github.com/computational-cell-analytics/micro-sam. The use of APG is documented at https://computational-cell-analytics.github.io/micro-sam/micro_sam.html#apg.

### 3.3. Datasets & metrics

We evaluate APG and other foundation models on 36 datasets from four different tasks and domains: nucleus segmentation in fluorescence microscopy (Greenwald et al., 2021; Ljosa et al., 2012; Arvidsson et al., 2023; Kromp et al., 2020; Vijayan et al., 2024; Zheng et al., 2023; Alwes et al., 2016; Caicedo et al., 2019; Mahbod et al., 2021b), cell segmentation in fluorescence microscopy (Stringer et al., 2021; Shi et al., 2025; Greenwald et al., 2022; Wolny et al., 2020; Pape et al., 2021; Ouyang et al., 2019; Willis et al., 2016; Bondarenko et al., 2023), cell segmentation in label-free microscopy (Edlund et al., 2021; Cutler et al., 2022; Spahn et al., 2022; Tsai et al., 2019; Vicar et al., 2021; Zargari et al., 2023; Seiffarth et al., 2024; Gupta et al., 2023; Dietler et al., 2020), and nucleus segmentation in histopathology (Wang et al., 2024; Naji et al., 2024; Kumar et al., 2019; Gamper et al., 2020; Naylor et al., 2018; Mahbod et al., 2024; Schuiveling et al., 2025; Vadori et al., 2025; Mahbod et al., 2021a). We use the test splits for all of these datasets. For volumetric datasets we run and evaluate the segmentation in 2D and sub-sample the test sets for efficiency reasons. Note that some of the methods we evaluate were trained on some of these datasets (though not only on the train splits, not on the test splits). See Fig. 2 for details. A detailed overview of all datasets can be found in Tab. 2.

We use the mean segmentation accuracy, following (Caicedo et al., 2019), to evaluate instance segmentation results. The mean segmentation accuracy (mSA) is based on true positives ($TP$), false negatives ($FN$), and false positives ($FP$), which are derived from the intersection over union (IoU) of predicted and true objects. Specifically, $TP(t)$ is defined as the number of matches between predicted and true objects with an IoU above the threshold $t$, $FP(t)$ correspond to the number of predicted objects minus $TP(t)$, and $FN(t)$ to the number of true objects minus $TP(t)$. The mean segmentation accuracy is computed over multiple thresholds:

$$\text{Mean Segmentation Accuracy} = \frac{1}{|\# \text{ thresholds}|} \sum_t \frac{TP(t)}{TP(t) + FP(t) + FN(t)}.$$

Here, we use thresholds $t \in [0.5, 0.55, 0.6, 0.65, 0.7, 0.75, 0.8, 0.85, 0.9, 0.95]$. For each dataset, we report the average mean segmentation accuracy over images in the test set. This metric is commonly used to evaluate instance segmentation in microscopy, see (Hirling et al., 2024) for an in-depth discussion.

### 4. Results

We evaluate SAM (w/ AMG), SAM3, $\mu$SAM (w/ AIS, APG), PathoSAM (w/ AIS, APG), CellPoseSAM, and CellSAM (see Sec. 3) on the data described in Sec. 3.3. We also evaluate CellPose3 (Stringer and Pachitariu, 2025), which uses a convolutional architecture and a smaller training set but is otherwise similar to CellPoseSAM. PathoSAM is only evaluated

on histopathology data and $\mu$SAM is not evaluated in this domain to account for the specific focus of these models. Note that we do not evaluate SAM2 (w/ AMG) as we found it to be inferior to SAM in this setting. It did not segment any objects for several datasets in initial experiments we ran. SAM3 is prompted with the single short noun phrase "cell" for all images. See Sec. 4.2 for a detailed analysis on the choice of text prompt.

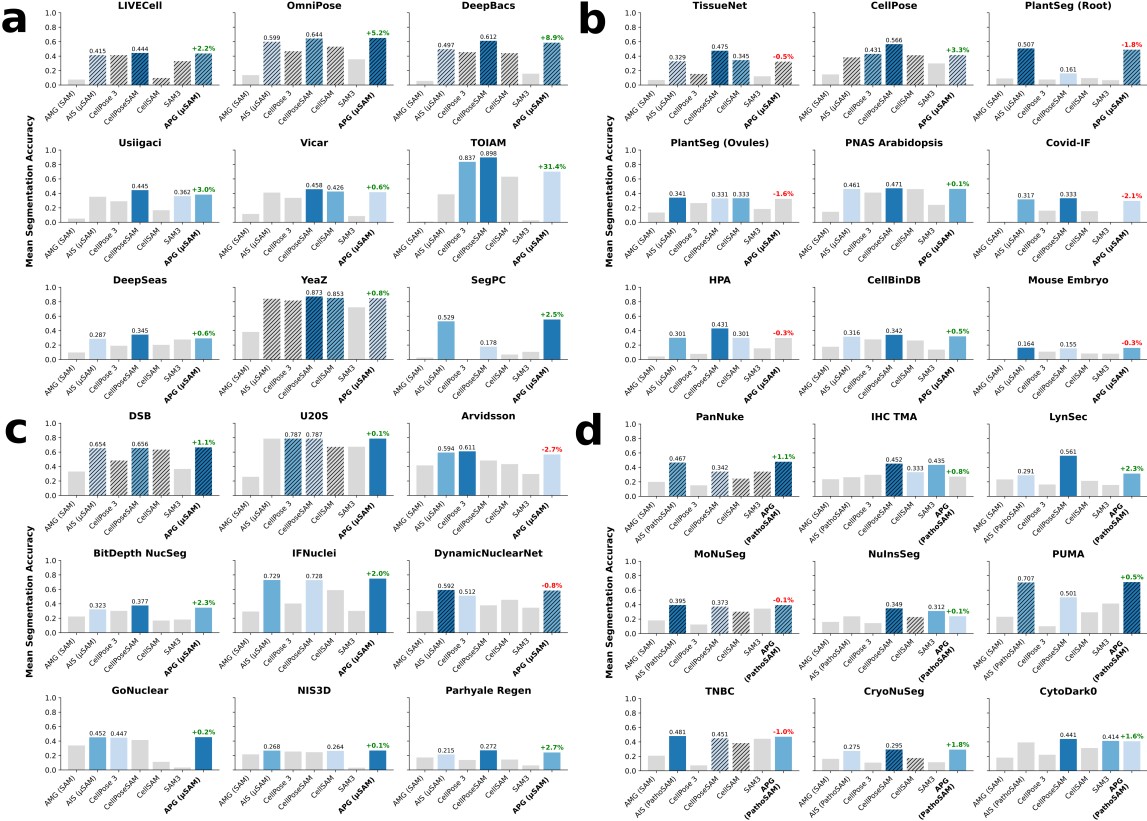

Figure 2: Results for 36 microscopy segmentation datasets in four different modalities: cells (fluorescence, a), cells (label-free, b), nuclei (fluorescence, c), and nuclei (histopathology, d). We indicate the top-3 ranked method with blue colors and methods that were trained on the corresponding training split with textured bars. For our method (APG) we indicate the absolute performance difference with respect to the reference method (AIS).

The result summary is shown in Fig. 1, the results for all datasets in Fig. 2, reported separately across the four imaging modalities. We highlight the three top performing methods and indicate whether the model was trained on the respective data's train split.

APG improves the $\mu$SAM model compared to AIS for all label-free microscopy cases, including a very substantial improvement for TOIAM. It improves 3 / 9 datasets for cell segmentation in fluorescence microscopy, with only modest differences in segmentation quality, improves 7 / 9 datasets for nucleus segmentation in fluorescence, and 7 / 9 for nucleus seg-

mentation in histopathology. Overall, we find that CellPoseSAM and APG are consistently among the best approaches, ranking among the top three for all four modalities. AIS is the third best model overall, followed by CellPose 3. CellSAM performs worse than the other microscopy foundation models, but consistently better than SAM and SAM3. Tables with all numerical results can be found in App. C. Qualitative results for selected methods and one dataset per domain are shown in Fig. 3 and for all datasets in Figs. 5 - 8. We further evaluate the statistical significance of differences between the models in App. D.

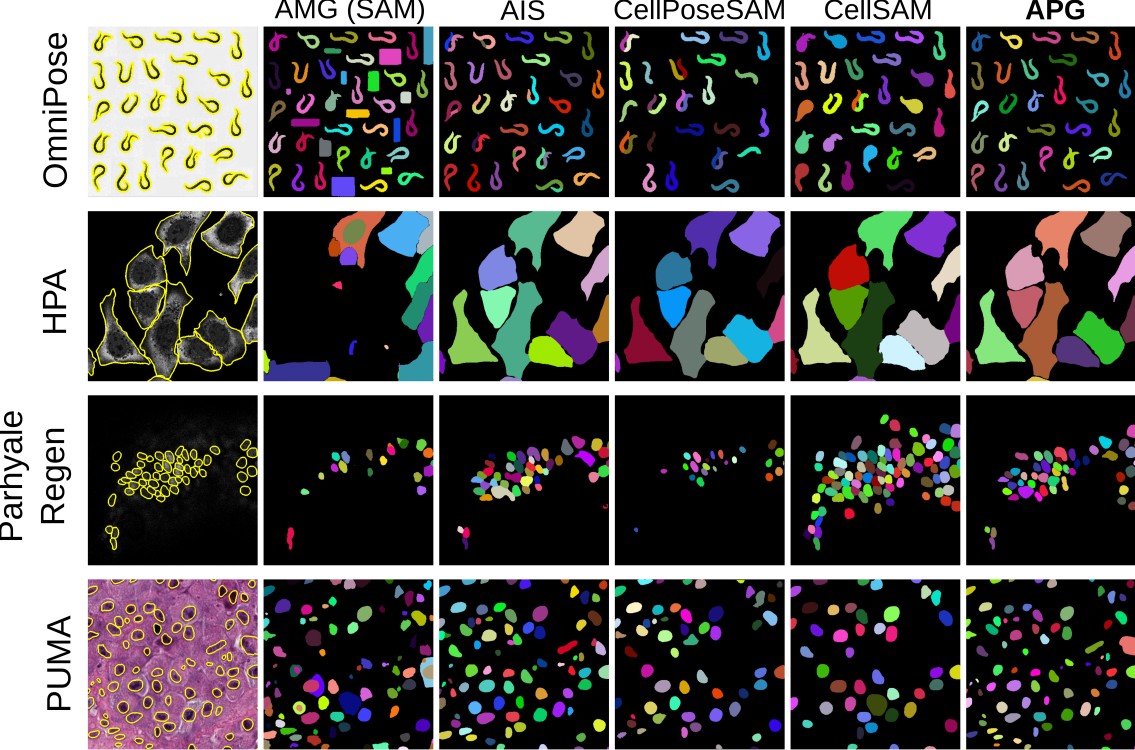

Figure 3: Qualitative segmentation results for all microscopy foundation models. We show examples for one dataset per domain / task: cell segmentation in label-free microscopy, cell segmentation in fluorescence microscopy, nucleus segmentation in fluorescence microscopy, and nucleus segmentation in histopathology (top to bottom). Examples for all datasets can be found in Figs. 5 - 8.

## 4.1. Comparison of APG strategies

We also compare an alternative strategy for deriving prompts in APG, using the foreground-restricted maxima of the boundary distance predictions. The corresponding results are shown in Fig. 4. The strategy using connected components, as described in Sec. 3.2, is superior.

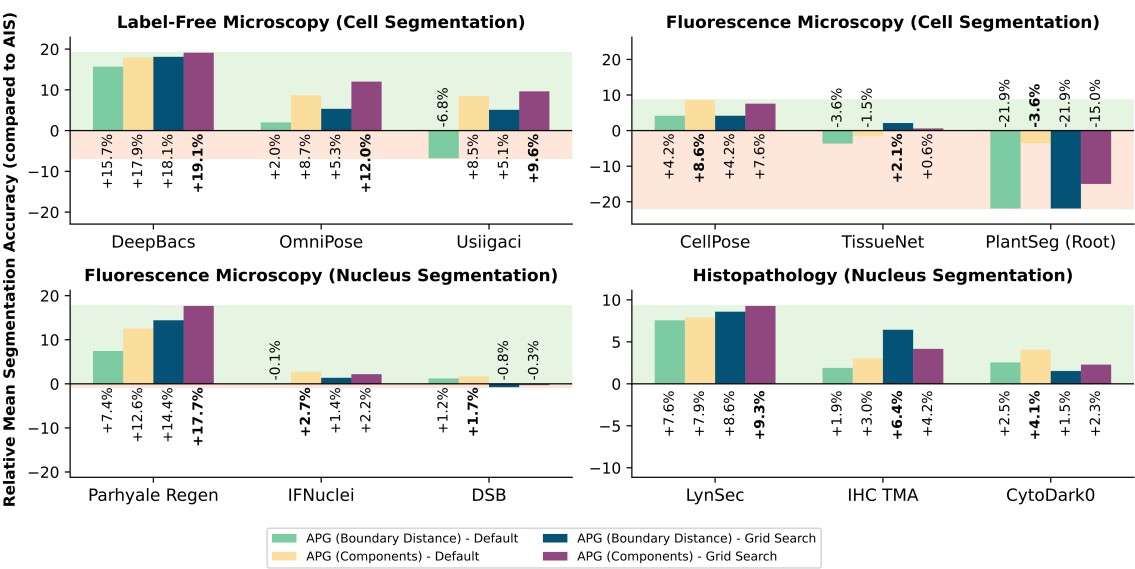

Figure 4: Comparison of our connected component-based strategy APG (APG (Components)) with an alternative deriving prompts from distance map maxima (APG (Boundary)) for the four different modalities (a-d), reporting the relative mean segmentation accuracy difference w.r.t AIS.

## 4.2. Prompting strategies for SAM3

The results for SAM3 in Fig. 2 were obtained by prompting the model with the phrase "cell". We perform an additional experiment to determine the sensitivity of SAM3 to the text prompt on four datasets (one per modality). We evaluate the segmentation quality with prompts corresponding to the correct biological term ("cell", "nucleus"), single nouns describing the object's shapes ("blob", "dot"), and combinations of adjectives and nouns describing the shapes. We evaluate both singular and plural for all cases. The results are shown in Tab. 1. They show that SAM3 lacks the knowledge of biological terms ("nucleus" is not recognized) and that it is overall fairly sensitive to the choice of prompts. For example, prompts describing the shape perform substantially better than the correct biological term "cell" in multiple cases. We did not yet prompt SAM3 with an example image and representative segmented objects.

| Dataset | LIVECell | | CellPose | | DSB | | PanNuke | |
|---|---|---|---|---|---|---|---|---|
| **Modality** | Label Free | | Fluorescence | | Fluorescence | | Histopathology | |
| **Task** | Cell | | Cell | | Nucleus | | Nucleus | |
| **Text Prompts** | | | | | | | | |
| cell(s) | ✓ (**0.331**) | ✓ (0.311) | ✓ (0.299) | ✓ (0.231) | ✓ (0.366) | ✓ (0.386) | ✓ (0.341) | ✓ (0.159) |
| nucleus (nuclei) | − | − | − | − | × | ✓ (0.085) | × | × |
| blob(s) | × | × | × | ✓ (0.175) | ✓ (0.103) | ✓ (0.416) | × | ✓ (0.179) |
| dot(s) | × | ✓ (0.014) | ✓ (0.015) | ✓ (0.096) | ✓ (0.047) | ✓ (0.387) | ✓ (0.007) | ✓ (0.021) |
| bright spot(s) | × | × | × | ✓ (0.027) | ✓ (0.465) | ✓ (**0.509**) | × | × |
| irregular shape(s) | ✓ (0.282) | ✓ (0.119) | ✓ (**0.301**) | ✓ (0.247) | ✓ (0.489) | ✓ (0.427) | ✓ (**0.379**) | ✓ (0.319) |
| large particle(s) | × | × | ✓ (0.005) | ✓ (0.002) | ✓ (0.199) | ✓ (0.076) | × | × |

Table 1: SAM3 results for text prompt-based microscopy segmentation. Datasets marked in italic font are part of SAM3's training data, datasets marked in bold font are not. The results for best performing text prompts per dataset are marked in **bold**. All reported scores are mean segmentation accuracy for the entire dataset. Prompts resulting in a score $<=0.001$ are marked as "×". Prompts that are not applicable to the given segmentation task are marked as "−".

## 5. Discussion

We introduced APG, a new method for instance segmentation with SAM-based foundation models. It consistently and substantially improved the segmentation results of $\mu$SAM and PathoSAM compared to the prior AIS approach and is competitive with the state-of-the-art CellPoseSAM. APG was applied without the need for re-training. It could also be applied to other SAM-based models, e.g. CellPoseSAM. However, in this case it would either rely on another model for prompt-based segmentation or joint training of CellPoseSAM with the objective for interactive segmentation.

Furthermore, we evaluated SAM3, the latest model of the SAM family, for microscopy. It performed well – though not yet competitive with domain-specific foundation models –, despite being trained on only two relevant datasets (Edlund et al., 2021; Gamper et al., 2020). Moreover, it showed a sensitivity to the choice of text prompt. Finetuning SAM3 on microscopy data promises to yield further improvements in this domain. We did not yet test prompting SAM3 with examples of annotated images, which could lead to substantial improvements without further training.

Overall, we saw a clear impact of model training data on performance, with models typically performing similar when trained on the respective data's training split, and more pronounced for "out-of-domain" datasets. Yet, we found that APG can lead to substantial improvements for both in-domain (e.g. OmniPose, DeepBacs) and out-of-domain (e.g. TOIAM) case. Overall, the performance of models seems to correlate with the size of their domain-specific training data, CellPose and $\mu$SAM (w/ AIS & APG) having the largest training sets and overall best performances. This observation also indicates that further finetuning to improve for specific applications remains relevant, as already shown in prior work (Teuber et al., 2025; Zhou et al., 2024) that demonstrated substantial improvements through training on small datasets – even a single image. These approaches would also directly translate to specifically improving APG.

Furthermore, APG could likely be improved by incorporating box prompts, which generally perform better than point prompts (Archit et al., 2025a). We performed initial experiments to derive candidate box prompts from the $\mu$SAM decoder predictions. However, we found that deriving a set of high-quality prompts that over-sample objects was challenging and did not yet find a strategy competitive with the simpler point prompt derivation. Future work, such as iterative prompt derivation, may enable such improvements.

Finally, a limitation of our study is the restriction to 2D evaluation, also for 3D datasets. CellPoseSAM, $\mu$SAM, and SAM3 support 3D segmentation, so this evaluation would provide a further valuable comparison and would further inform how to improve microscopy foundation models. We plan to address this problem in future work, potentially including adaptations of SAM2 and/or SAM3 to microscopy.

## Acknowledgments

The work of Anwai Archit was funded by the Deutsche Forschungsgemeinschaft (DFG, German Research Foundation) - PA 4341/2-1. Constantin Pape is supported by the German Research Foundation (Deutsche Forschungsgemeinschaft, DFG) under Germany's Excellence Strategy - EXC 2067/1-390729940. This work is supported by the Ministry of Science and Culture of Lower Saxony through funds from the program zukunft.niedersachsen of the Volkswagen Foundation for the 'CAIMed – Lower Saxony Center for Artificial Intelligence and Causal Methods in Medicine' project (grant no. ZN4257). It was also supported by the Google Research Scholarship "Vision Foundation Models for Bioimage Segmentation". We gratefully acknowledge the computing time granted by the Resource Allocation Board and provided on the supercomputer Emmy at NHR@Göttingen as part of the NHR infrastructure, under the project nim00007. We would like to thank Sebastian von Haaren for suggestions on data visualizations, Carolin Teuber and Titus Griebel for data processing scripts, and Julia Jeremias for post-processing scripts.

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

# Appendix A. Dataset Details

| Dataset | Imaging Modality | Input Dimensions |
|---|---|---|
| LIVECell (Edlund et al., 2021) | Phase Contrast | 2D |
| OmniPose (Cutler et al., 2022) | Phase Contrast & Brightfield | 2D |
| DeepBacs (Spahn et al., 2022) | Brightfield & Fluorescence | 2D |
| Usiigaci (Tsai et al., 2019) | Phase Contrast | 2D |
| Vicar (Vicar et al., 2021) | Quantitative Phase | 2D |
| TOIAM (Seiffarth et al., 2024) | Phase Contrast | 2D+T |
| DeepSeas (Zargari et al., 2023) | Phase Contrast | 2D |
| YeaZ (Dietler et al., 2020) | Brightfield & Phase Contrast | 2D & 2D+T |
| SegPC (Gupta et al., 2023) | Brightfield | 2D |
| TissueNet (Greenwald et al., 2022) | Fluorescence | 2D |
| CellPose (Stringer et al., 2021) | Fluorescence | 2D |
| PlantSeg (Root) (Wolny et al., 2020) | Light-Sheet Fluorescence | 3D |
| PlantSeg (Ovules) (Wolny et al., 2020) | Confocal | 3D |
| PNAS Arabidopsis (Willis et al., 2016) | Confocal | 3D |
| Covid-IF (Pape et al., 2021) | Immunofluorescence | 2D |
| HPA (Ouyang et al., 2019) | Confocal | 2D |
| CellBinDB (Shi et al., 2025) | Multiple | 2D |
| Mouse Embryo (Bondarenko et al., 2023) | Confocal | 3D |
| DSB (Caicedo et al., 2019) | Fluorescence | 2D |
| U20S (Ljosa et al., 2012) | Fluorescence | 2D |
| Arvidsson (Arvidsson et al., 2023) | High-Content Fluorescence | 2D |
| BitDepth NucSeg (Mahbod et al., 2021b) | Fluorescence | 2D |
| IFNuclei (Kromp et al., 2020) | (Immuno)Fluorescence | 2D |
| DynamicNuclearNet (Greenwald et al., 2021) | Fluorescence | 2D+T |
| GoNuclear (Vijayan et al., 2024) | Fluorescence | 3D |
| NIS3D (Zheng et al., 2023) | Light-Sheet Fluorescence | 3D |
| Parhyale Regen (Alwes et al., 2016) | Confocal | 3D |
| PanNuke (Gamper et al., 2020) | *H&E staining* | 2D |
| IHC TMA (Wang et al., 2024) | *IHC staining* | 2D |
| LynSec (Naji et al., 2024) | *IHC staining* | 2D |
| MoNuSeg (Kumar et al., 2019) | *H&E staining* | 2D |
| NuInsSeg (Mahbod et al., 2024) | *H&E staining* | 2D |
| PUMA (Schuiveling et al., 2025) | *H&E staining* | 2D |
| TNBC (Naylor et al., 2018) | *H&E staining* | 2D |
| CryoNuSeg (Mahbod et al., 2021a) | *(Cryo-Sectioned) H&E staining* | 2D |
| CytoDark0 (Vadori et al., 2025) | *Nissl staining* | 2D |

Table 2: Description of the different datasets used in our study. For the 3D datasets / 2D+T datasets, we evaluate over individual slices / frames. For the histopathology datasets, the imaging modality describes the staining protocol (in italics) for the images.

# Appendix B. Extended Qualitative Results

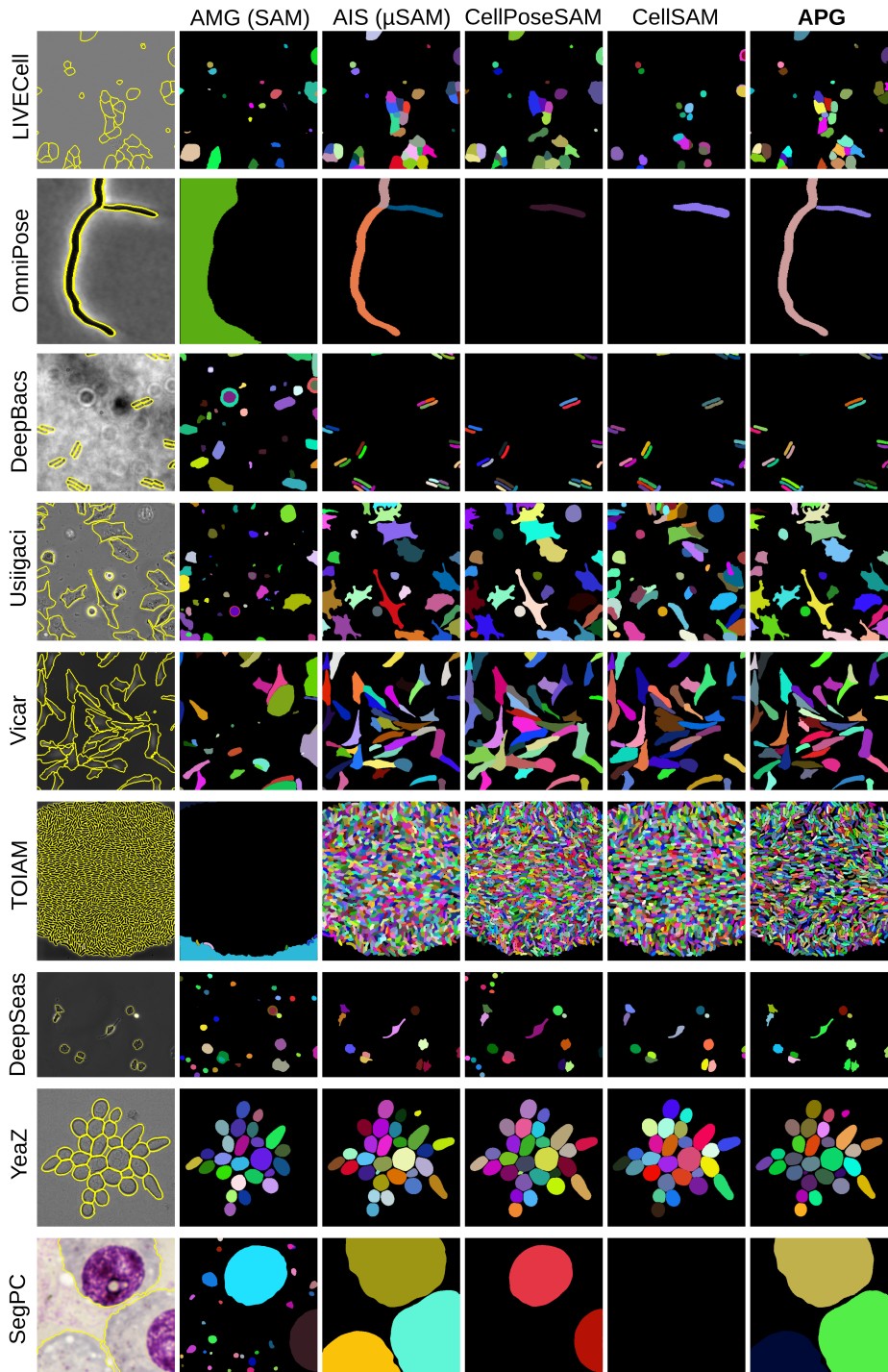

Figure 5: Qualitative results for all label-free microscopy datasets for cell instance segmentation.

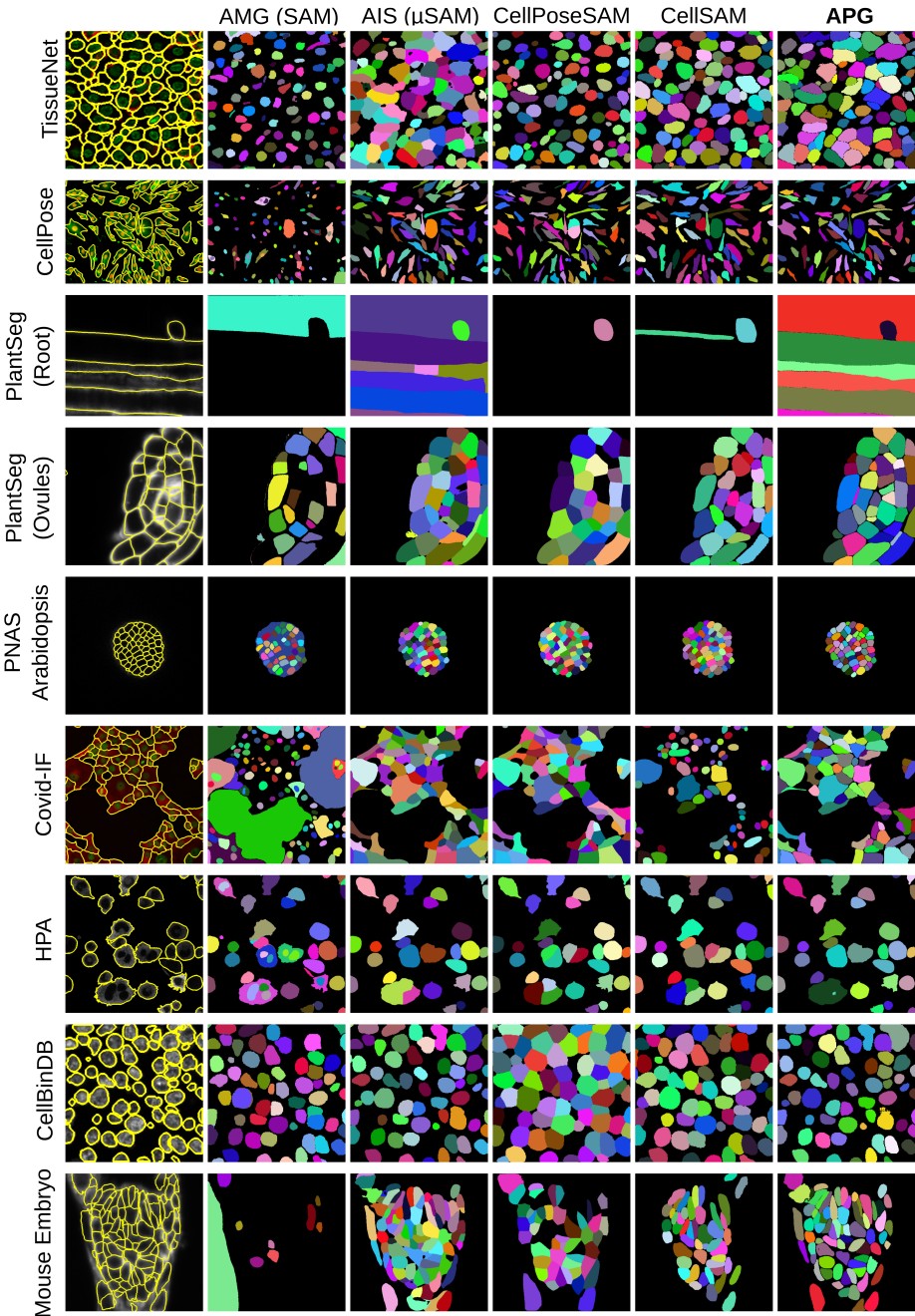

Figure 6: Qualitative results for all fluorescence microscopy datasets for cell instance segmentation.

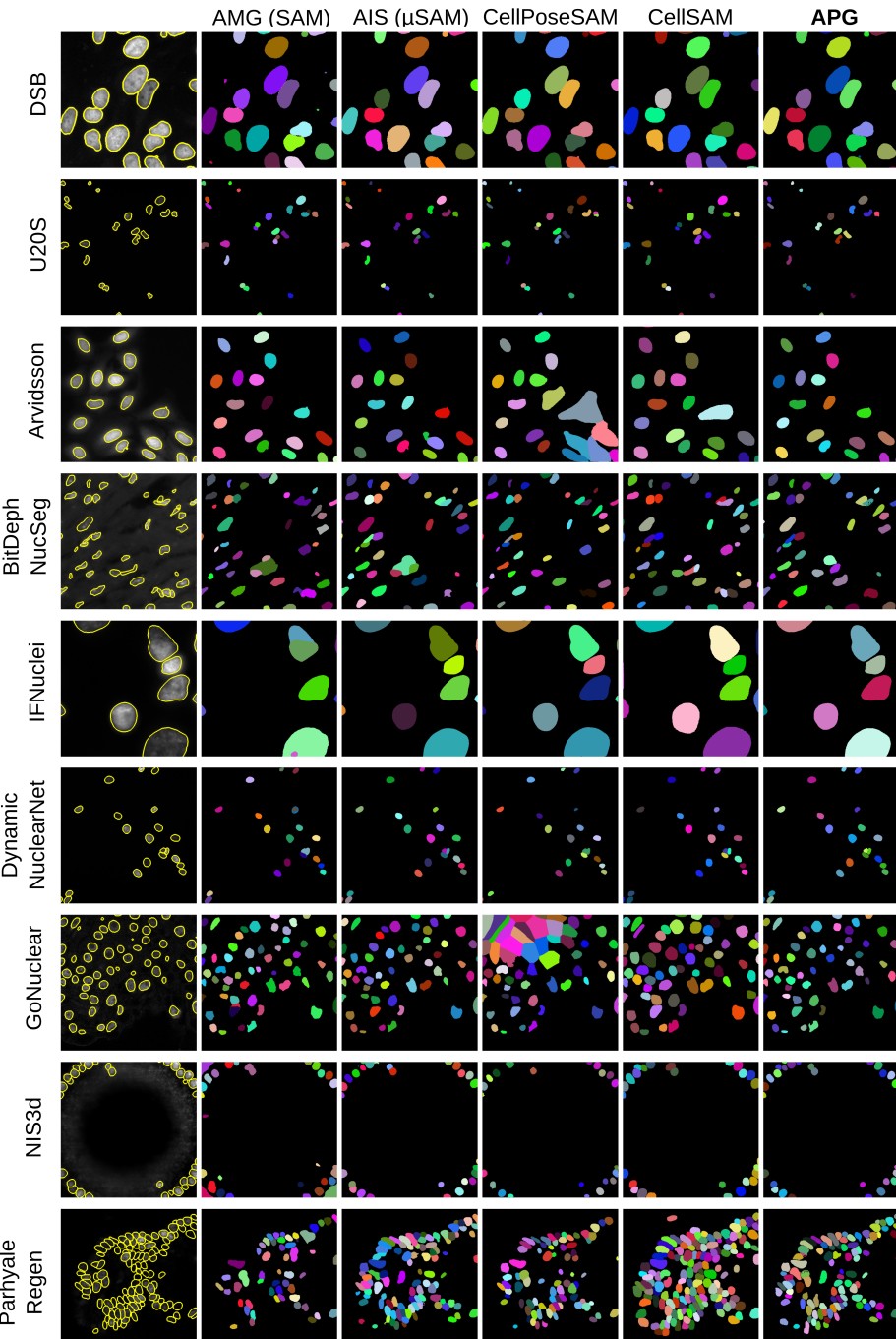

Figure 7: Qualitative results for all fluorescence microscopy datasets for nucleus instance segmentation.

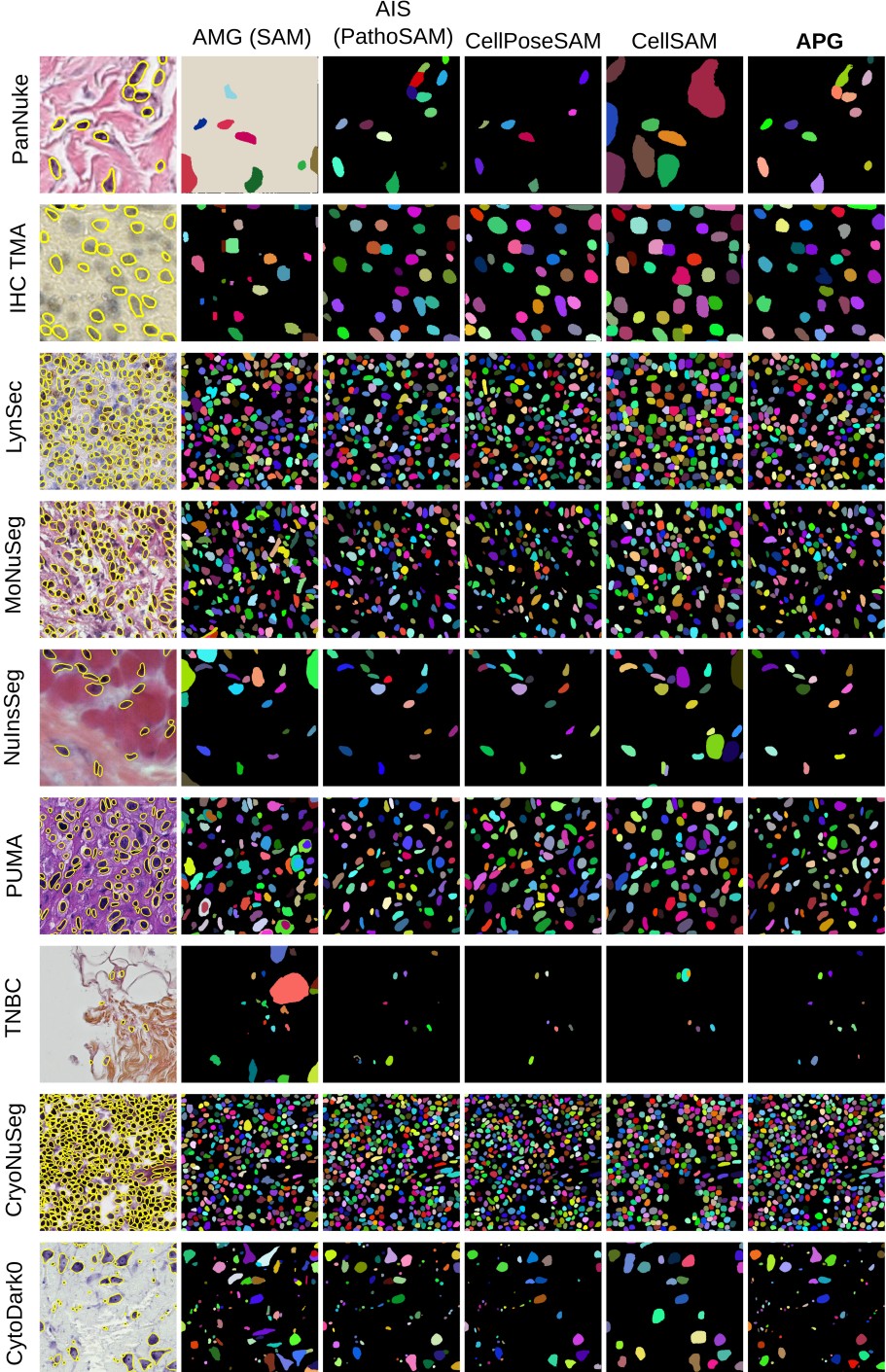

Figure 8: Qualitative results for all histopathology datasets for nucleus instance segmentation.

## Appendix C. Quantitative Results

Detailed quantitative results (both mean over all datasets and per dataset (per domain)).

| Method | Label-Free (Cell) | Fluorescence (Cell) | Fluorescence (Nucleus) | Histopathology (Nucleus) |
|---|---|---|---|---|
| AMG (SAM) | 0.106 | 0.091 | 0.283 | 0.200 |
| AIS | *0.480* | 0.347 | 0.513 | *0.390* |
| SAM3 | 0.269 | 0.143 | 0.255 | 0.331 |
| CellPose 3 | 0.424 | 0.218 | 0.438 | 0.155 |
| CellPoseSAM | **0.544** | **0.363** | *0.483* | **0.418** |
| CellSAM | 0.380 | 0.272 | 0.386 | 0.277 |
| **APG** | 0.541 | *0.344* | **0.518** | 0.398 |

Table 3: Mean Segmentation Accuracy (mSA) averaged over all datasets for each modality. The best / second / third ranking methods are shown in **bold** / underline / *italics*. For histopathology, AIS / APG correspond to the PathoSAM, and $\mu$SAM for others.

| Dataset | AMG (SAM) | AIS ($\mu$SAM) | SAM3 | CellPose 3 | CellPoseSAM | CellSAM | APG ($\mu$SAM) |
|---|---|---|---|---|---|---|---|
| LIVECell | 0.075 | *0.415* | 0.331 | 0.414 | **0.444** | 0.098 | 0.437 |
| OmniPose | 0.137 | *0.599* | 0.356 | 0.468 | 0.644 | 0.531 | **0.651** |
| DeepBacs | 0.057 | *0.497* | 0.157 | 0.455 | **0.612** | 0.441 | 0.586 |
| Usiigaci | 0.051 | 0.353 | *0.362* | 0.291 | **0.445** | 0.167 | 0.383 |
| Vicar | 0.115 | 0.411 | 0.086 | 0.338 | **0.458** | 0.426 | *0.417* |
| TOIAM | 0.009 | 0.387 | 0.027 | 0.837 | **0.898** | 0.631 | *0.701* |
| DeepSeas | 0.098 | *0.287* | 0.277 | 0.191 | **0.345** | 0.203 | 0.293 |
| YeaZ | 0.382 | 0.841 | 0.723 | 0.817 | **0.873** | 0.853 | *0.849* |
| SegPC | 0.027 | 0.529 | 0.106 | 0.001 | *0.178* | 0.069 | **0.554** |

Table 4: Quantitative results on label-free microscopy datasets for cell segmentation: mean segmentation accuracy (mSA) per dataset and method. For each dataset, the best / second / third ranking methods are shown in **bold** / underline / *italics*.

| Dataset | AMG (SAM) | AIS (μSAM) | SAM3 | CellPose 3 | CellPoseSAM | CellSAM | APG (μSAM) |
|---|---|---|---|---|---|---|---|
| TissueNet | 0.069 | *0.329* | 0.121 | 0.154 | **0.475** | 0.345 | 0.324 |
| CellPose | 0.147 | 0.383 | 0.299 | 0.431 | **0.566** | 0.413 | *0.416* |
| PlantSeg (Root) | 0.091 | **0.507** | 0.067 | 0.076 | *0.161* | 0.096 | 0.489 |
| PlantSeg (Ovules) | 0.135 | **0.341** | 0.184 | 0.266 | *0.331* | 0.333 | 0.325 |
| PNAS Arabidopsis | 0.145 | *0.461* | 0.241 | 0.411 | **0.471** | 0.459 | 0.462 |
| Covid-IF | 0.007 | 0.317 | 0.005 | 0.161 | **0.333** | 0.154 | *0.296* |
| HPA | 0.043 | 0.301 | 0.155 | 0.078 | **0.431** | 0.301 | *0.298* |
| CellBinDB | 0.177 | *0.316* | 0.137 | 0.279 | **0.342** | 0.264 | 0.321 |
| Mouse Embryo | 0.003 | **0.164** | 0.081 | 0.109 | *0.155* | 0.083 | 0.161 |

Table 5: Quantitative results on fluorescence microscopy datasets for cell segmentation: mean segmentation accuracy per dataset and method. For each dataset, the best / second / third ranking methods are shown in **bold** / underline / *italics*.

| Dataset | AMG (SAM) | AIS (μSAM) | SAM3 | CellPose 3 | CellPoseSAM | CellSAM | APG (μSAM) |
|---|---|---|---|---|---|---|---|
| DSB | 0.331 | *0.654* | 0.367 | 0.484 | 0.656 | 0.634 | **0.665** |
| U20S | 0.258 | 0.786 | *0.674* | **0.787** | **0.787** | 0.673 | **0.787** |
| Arvidsson | 0.416 | 0.594 | 0.297 | **0.611** | 0.484 | 0.434 | *0.567* |
| BitDepth NucSeg | 0.224 | *0.323* | 0.182 | 0.302 | **0.377** | 0.168 | 0.346 |
| IFNuclei | 0.293 | 0.729 | 0.301 | 0.404 | *0.728* | 0.589 | **0.749** |
| Dynamic-NuclearNet | 0.298 | **0.592** | 0.346 | *0.512* | 0.379 | 0.455 | 0.584 |
| GoNuclear | 0.339 | 0.452 | 0.034 | *0.447* | 0.415 | 0.112 | **0.454** |
| NIS3D | 0.216 | 0.268 | 0.031 | 0.255 | 0.246 | *0.264* | **0.269** |
| Parhyale Regen | 0.173 | *0.215* | 0.063 | 0.138 | **0.272** | 0.144 | 0.242 |

Table 6: Quantitative results on fluorescence microscopy datasets for nucleus segmentation: mean segmentation accuracy per dataset and method. For each dataset, the best / second / third ranking methods are shown in **bold** / underline / *italics*.

| Dataset | AMG (SAM) | AIS (PathoSAM) | SAM3 | CellPose 3 | CellPoseSAM | CellSAM | APG (PathoSAM) |
|---|---|---|---|---|---|---|---|
| PanNuke | 0.199 | 0.467 | 0.341 | 0.152 | *0.342* | 0.244 | **0.478** |
| IHC TMA | 0.236 | 0.264 | 0.435 | 0.297 | **0.452** | *0.333* | 0.272 |
| LynSec | 0.233 | *0.291* | 0.157 | 0.163 | **0.561** | 0.213 | 0.314 |
| MoNuSeg | 0.182 | **0.395** | 0.345 | 0.125 | *0.373* | 0.302 | 0.394 |
| NuInsSeg | 0.161 | 0.238 | 0.312 | 0.144 | **0.349** | 0.229 | *0.239* |
| PUMA | 0.232 | 0.707 | 0.415 | 0.101 | *0.501* | 0.294 | **0.712** |
| TNBC | 0.209 | **0.481** | 0.443 | 0.075 | *0.451* | 0.383 | 0.471 |
| CryoNuSeg | 0.165 | *0.275* | 0.118 | 0.113 | **0.295** | 0.177 | 0.293 |
| CytoDark0 | 0.182 | 0.393 | 0.414 | 0.222 | **0.441** | 0.315 | *0.409* |

Table 7: Quantitative results on histopathology datasets for nucleus segmentation: mean segmentation accuracy per dataset and method. For each dataset, the best / second / third ranking methods are shown in **bold** / underline / *italics*.

## Appendix D. Statistical Evaluation

We additionally performed paired Wilcoxon signed-rank tests on per-image mSA differences for all method pairs. Figs. 9 − 12 summarize the resulting wins, losses, and draws across the four imaging modalities. Overall, the statistical analysis confirms the main ranking trends, with large score differences being consistently significant and the relative ordering of the strongest methods remaining broadly unchanged.

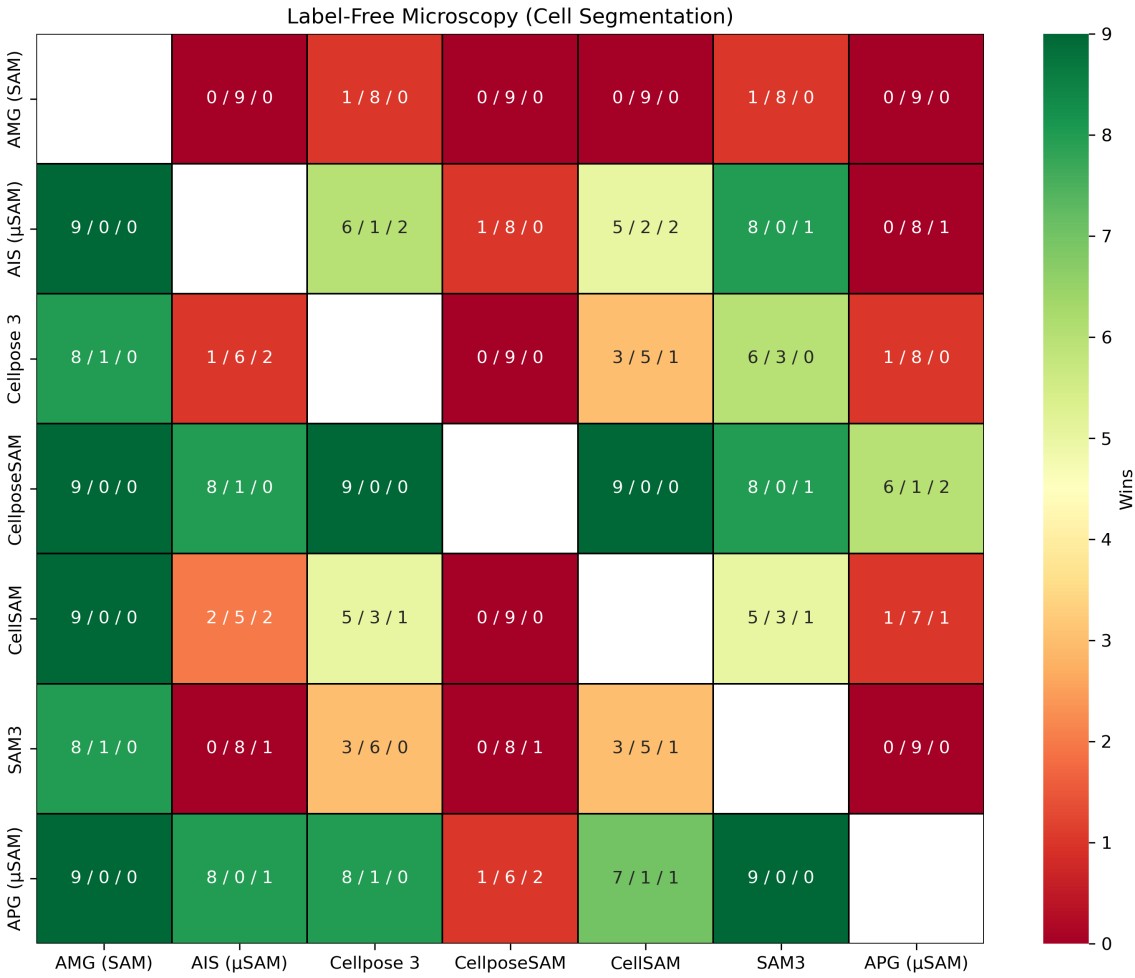

Figure 9: Statistical evaluation results for all label-free microscopy datasets for cell instance segmentation.

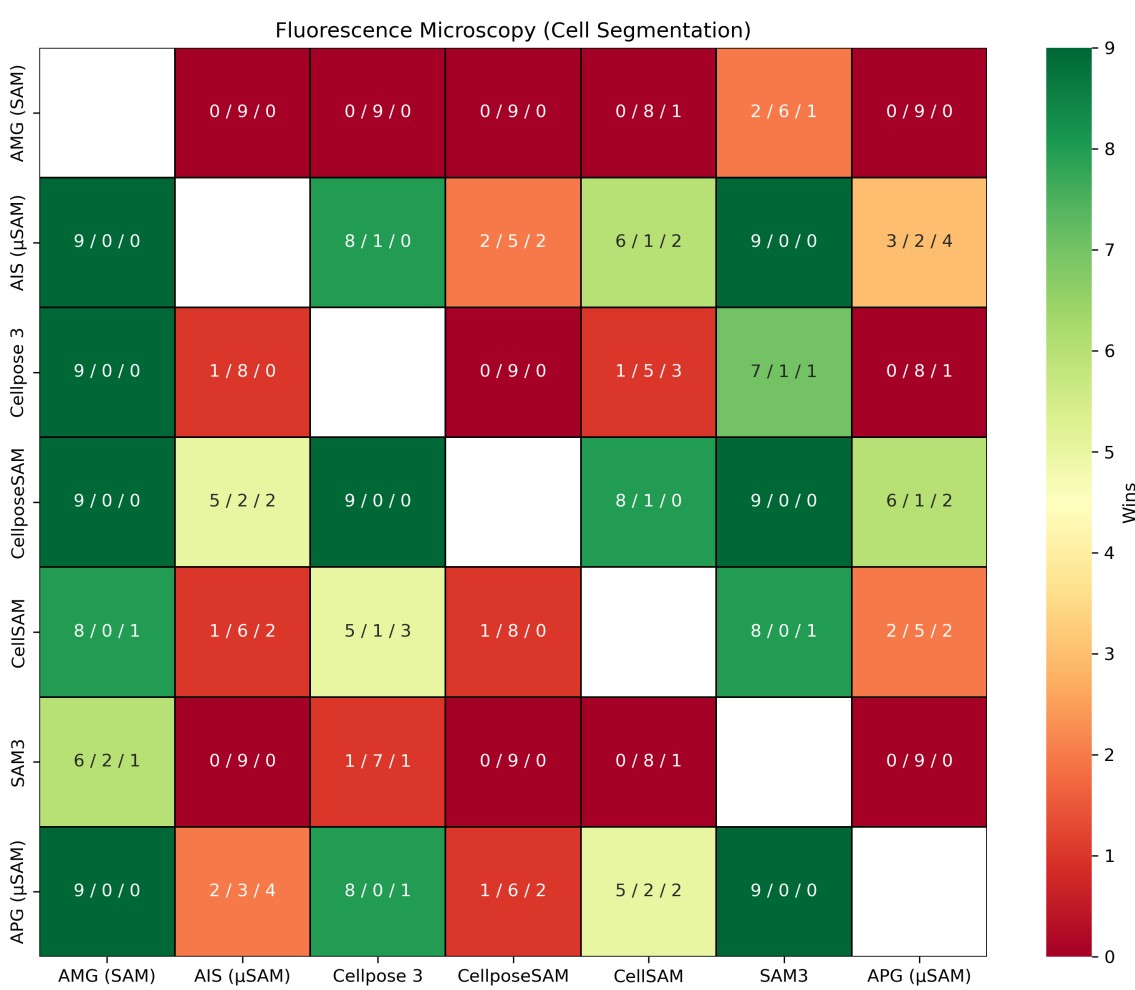

Figure 10: Statistical evaluation results for all fluorescence microscopy datasets for cell instance segmentation.

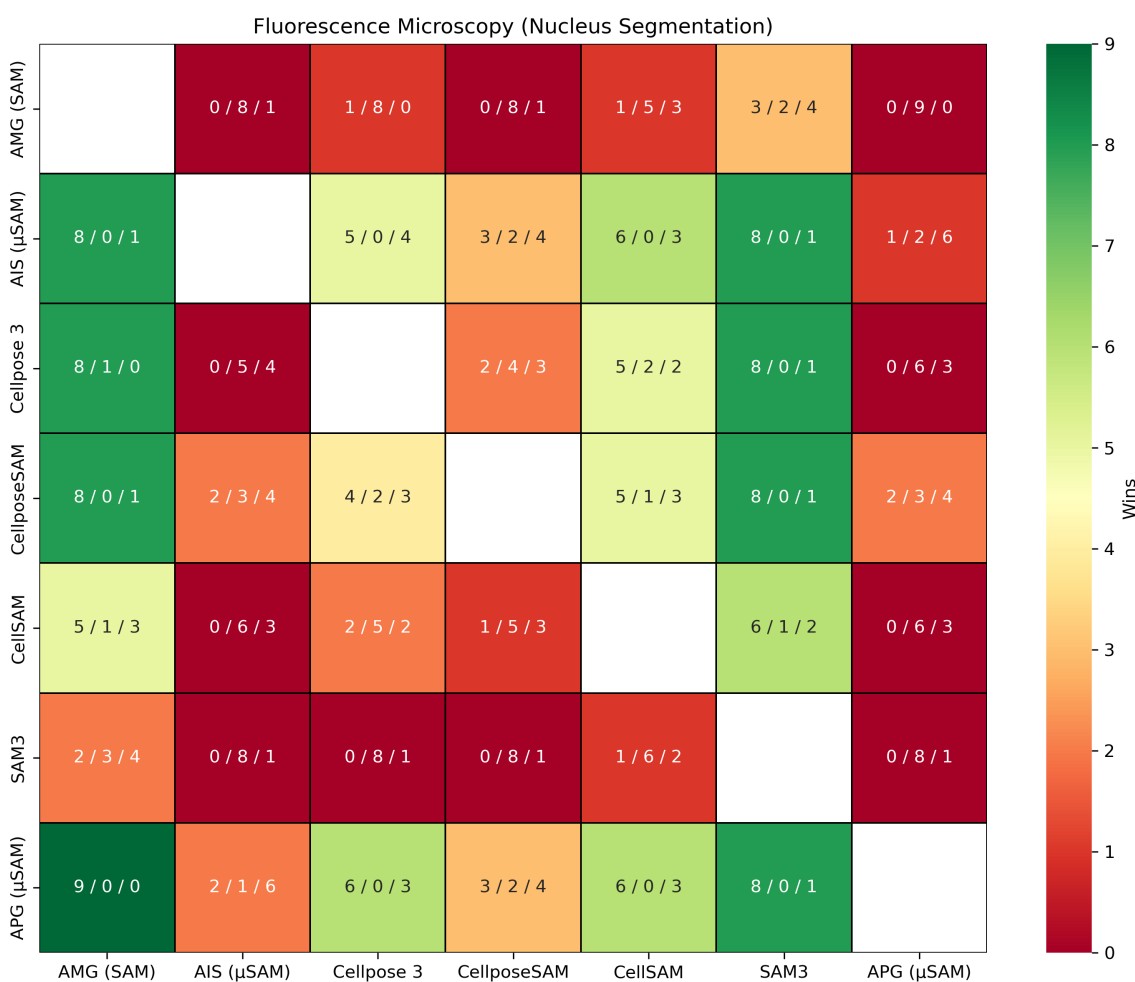

Figure 11: Statistical evaluation results for all fluorescence microscopy datasets for nucleus instance segmentation.

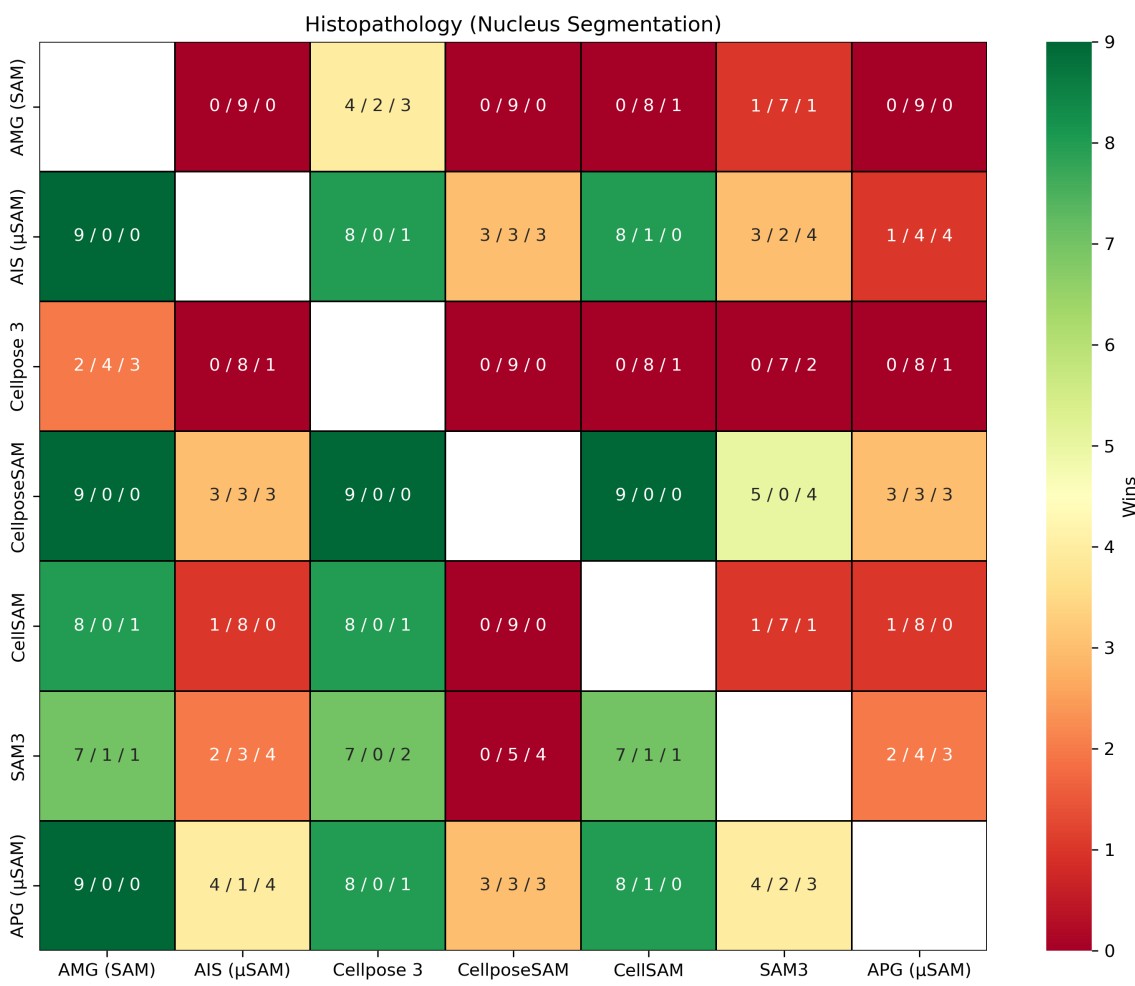

Figure 12: Statistical evaluation results for all histopathology datasets for nucleus instance segmentation.

