# OpenReview forum: "Revisiting foundation models for cell instance segmentation"
_MIDL.io/2026/Conference — MIDL 2026 Poster_

### Official Review · Reviewer_b2EZ · 2026-01-08

**Confidence:** 4
**Preliminary Rating:** 4
**Final Rating:** 4

**Summary:**

The authors introduce Automatic Prompt Generation (APG), to enhance microscopy image segmentation with the Segment Anything Model (SAM). APG generates point prompts from the model's decoder predictions and applies NMS to select overlapping masks.
The method was thoroughly tested against general-purpose models (SAM, SAM2, SAM3) and domain specific foundation models (CellPose, SAM, CellSAM, SAM) on 36 diverse microscopy datasets involving cell, nucleus, and organoid segmentation tasks. Results from
the experiments show that APG delivers SOTA results, always being in the top three methods across the four data domains, and it is way ahead of the general purpose models such as SAM3 which have not yet developed domain specific vocabulary.

**Strengths:**

- Novelty: Introduction of Automatic Prompt Generation (APG) that can improve upon automatic instance segmentation without prompting.
- Comprehensive Benchmarking: The researchers evaluated a massive set of 36 datasets across four distinct domains: fluorescence cell segmentation, label-free cell segmentation, fluorescence nucleus segmentation, and histopathology.
- This work also comprehensively evaluates the recently published SAM3 on microscopy data.
- The paper discusses limitations, such as the current restriction to 2D evaluation for 3D datasets and the sensitivity of SAM3 to specific noun prompts.
- The paper is also well written with a clear structure and is easy to understand. They also do a very comprehensive survey of existing literature and previous work.

**Weaknesses:**

- The evaluation of SAM3 reveals a significant weakness in the model's reliance on text prompts. The authors acknowledge that the model's performance is highly dependent on specific noun prompts (e.g.,"cells," "nuclei"). This sensitivity suggests that the comparison between SAM3 and domain-specific models might be influenced more by prompts than architectural capabilities.
- Experimental details: No experimental details were provided in the paper. This is a major drawback for reproducibility. There is also no repository / codebase provided, which further reduces reproducibility.
- Data presentation: No tables were present. It would be better to compare the results if numbers were present.
- The study focuses exclusively on 2D segmentation. Given that many of the 36 datasets used (such as the NIS3D or organoid datasets) contain 3D images, evaluating them as 2D slices ignores the spatial consistency required in modern volumetric microscopy.

**Detailed Comments:**

- In Figure 2, the axis descriptions are not readable on a printout.
- I would generally recommend to use vector graphics in plots to avoid rastering artifacts such as in Figure 2.

**Justification Of Final Rating:**

I think this paper has significantly improved in the rebuttal phase. To summarize:

- The ablation / dependency study on the text prompt adds value to the manuscript.
- The code base is now available, reducing reproducibility issues.
- Figure texts are now vectorized, resolving problems with rasterized images, and have increased font sizes.

**Justification Of The Preliminary Rating:**

It's an overall nice paper about a relevant topic featuring an interesting approach with an extensive evaluation that also features an in-depth discussion. However, there are some non-negligible weaknesses, as outlined above, that should be fixed.

**Questions To Address In The Rebuttal:**

- The authors should explain the robustness of the APG (Automatic Prompt Generation) method across the 36 datasets. Since a single set of default parameters (tfg = 0.5; tb, tc, s, and tnms are parameters with default values 0.5, 0.5, 25, and 0.9) was used for all experiments, were these values optimized for a specific domain?
- The authors should provide the text prompts for SAM3
- The authors should list experimental details including hyperparameters
- Please either publish your code or provide a rationale why this is not possible.
- Please think about representing the bar plots in a numeric format that allows true comparison and also reproducibility for later work. You might provide the actual numbers in your repository or the appendix.

---

> ### Author Response · Authors · 2026-01-24
>
> Thank you very much for your careful review and the positive appraisal of our work. We have revised our manuscript based on your and the other reviewer’s feedback.
>
> Regarding the specific points you brought up:
>
> > The evaluation of SAM3 reveals a significant weakness in the model's reliance on text prompts. The authors acknowledge that the model's performance is highly dependent on specific noun prompts (e.g.,"cells," "nuclei"). This sensitivity suggests that the comparison between SAM3 and domain-specific models might be influenced more by prompts than architectural capabilities.
>
> This observation is correct. We now investigate the dependence of SAM3 on the text prompt explicitly by testing several prompt choices on four representative datasets. See the newly added Table 1. The fact that SAM3’s performance is contingent on the choice of text prompt is not in itself a weakness of our paper. Our goal is merely to evaluate and compare this model in the context of microscopy segmentation.
>
> > Experimental details: No experimental details were provided in the paper. This is a major drawback for reproducibility. There is also no repository / codebase provided, which further reduces reproducibility.
>
> It is unclear to us which experimental details are missing, the Methods and Results section provide the necessary information to understand the methodology and experiments. Furthermore, we have added Table 2 with a more detailed overview of the datasets used in our study. We have indeed forgotten a link to the code in the initial version of the manuscript and regret this oversight. It is available at https://github.com/computational-cell-analytics/micro-sam with details on how to use the APG functionality and reproduce our experiments provided at https://computational-cell-analytics.github.io/micro-sam/micro_sam.html#apg. We have added these links to the revised manuscript.
>
> > Data presentation: No tables were present. It would be better to compare the results if numbers were present.
>
> We have added Tables 3-7 with all numeric results to the Appendix.
>
> > The study focuses exclusively on 2D segmentation. Given that many of the 36 datasets used (such as the NIS3D or organoid datasets) contain 3D images, evaluating them as 2D slices ignores the spatial consistency required in modern volumetric microscopy.
>
> Extending our study to 3D segmentation is out of scope. This would substantially increase the complexity of experiments and obscure the comparison of the underlying 2D segmentation functionality due to differences in the 3D segmentation methodology between different methods. We agree that this is a worthwhile goal for future research and already mention this as a limitation of our study.
>
> >  In Figure 2, the axis descriptions are not readable on a printout. I would generally recommend to use vector graphics in plots to avoid rastering artifacts such as in Figure 2.
>
> Thank you for this comment. We have increased the font sizes and now use vector graphics in all the plots.
>
> > The authors should explain the robustness of the APG (Automatic Prompt Generation) method across the 36 datasets. Since a single set of default parameters (tfg = 0.5; tb, tc, s, and tnms are parameters with default values 0.5, 0.5, 25, and 0.9) was used for all experiments, were these values optimized for a specific domain?
>
> The default values are used for all experiments without any parameter tuning. This was already stated in the manuscript: “Here, we set $t_{fg}$ = 0.5; $t_b$, $t_c$, s, and $t_{nms}$ are parameters with default values 0.5, 0.5, 25, and 0.9. We run all experiments with these defaults” (From Section 3.2). We have slightly rephrased this text to make it even more clear.
>
> > The authors should provide the text prompts for SAM3.
>
> We use the prompt “cell” for the main experiments. We now state this more clearly in the text. See the newly added Table 1 for an analysis of SAM3’s performance depending on the choice of prompts.
>
> > The authors should list experimental details including hyperparameters
>
> All relevant parameters are already listed in Section 3.2. Note that we do not train any models ourselves, so there are no training hyperparameters.
>
> > Please either publish your code or provide a rationale why this is not possible.
>
> Our code is published and we now provide the link in the manuscript; see above.
>
> > Please think about representing the bar plots in a numeric format that allows true comparison and also reproducibility for later work. You might provide the actual numbers in your repository or the appendix.
>
> We have added a table with all results, see Tables 3-7 (Appendix).

---

> > ### Comment · Reviewer_b2EZ · 2026-01-29
> >
> > I think this paper has significantly improved in the rebuttal phase. To summarize:
> > - The ablation / dependency study on the text prompt adds value to the manuscript.
> > - The code base is now available, reducing reproducibility issues.
> > - Figure texts are now vectorized, resolving problems with rasterized images, and have increased font sizes.
> > - All results are now available numerically.

---

### Official Review · Reviewer_LmaQ · 2026-01-09

**Confidence:** 4
**Preliminary Rating:** 3
**Final Rating:** 4

**Summary:**

This paper presents a large-scale benchmark of SAM-based foundation models for microscopy instance segmentation and proposes Automatic Prompt Generation (APG) as a post-hoc strategy to improve segmentation quality without retraining. APG generates point prompts from the decoder predictions of µSAM and PathoSAM, which are then fed into SAM’s promptable mask decoder, followed by non-maximum suppression to produce final instance masks. The approach is evaluated on 36 microscopy datasets spanning cell, nucleus, and organoid segmentation across multiple imaging modalities.

**Strengths:**

- Extensive benchmarking across datasets and modalities.
- Lightweight, training-free improvement strategy.
- Clear empirical gains over automatic instance segmentation in many settings.
- The benchmarking is systematic and well design.

**Weaknesses:**

- The paper is clearly lack of statistical significance analysis.
- Figures are super dense and difficult to interpret.
- APG methodology (Section 3.2) is very hard to follow and insufficiently motivated, this would lead to limited novelty relative to prior work.
- Lack of qualitative results and visual validation, i.e., ambiguity among cell, nucleus and organoid segmentation claims.

**Detailed Comments:**

- Section 3.2 would benefit from an additional conceptual diagram or a simplified explanation of APG that emphasizes intuition over procedural detail.
- Key parameters should be more clearly defined, including whether they are fixed globally or tuned per dataset, and how sensitive performance is to these choices.
- Adding qualitative examples across different modalities (e.g., cells vs. nuclei, fluorescence vs. label-free imaging) would substantially strengthen the presentation.
- Some performance claims rely primarily on relative rankings rather than absolute performance differences, which may obscure small but practically meaningful gaps.

**Justification Of Final Rating:**

The rebuttal clearly improves clarity and presentation, few limitations remain that constrain overall confidence.

- The evaluation relies on rank-based comparisons rather than statistical significance testing or effect-size analysis, which limits insight into the practical relevance of the reported differences.

- The APG methodology is better motivated, yet the contribution remains primarily integrative, with limited evidence that it represents a conceptually distinct advance over prior approaches.

**Justification Of The Preliminary Rating:**

This paper provides a useful benchmark of SAM-based foundation models for microscopy and introduces a practical approach to improve segmentation performance without retraining. However, the contribution is largely incremental, the methodological description lacks clarity in key aspects, and the evaluation is weakened by the absence of statistical significance testing and sufficiently detailed qualitative validation.

**Questions To Address In The Rebuttal:**

Please check the Weakness and Detailed Comments section.

---

> ### Author Response · Authors · 2026-01-24
>
> Thank you very much for your review and of our work. We have revised our manuscript based on your and the other reviewer’s feedback.
>
> Regarding the specific points you brought up:
>
> > The paper is clearly lack of statistical significance analysis.
>
> Please note that we are comparing deterministic instance segmentation methods without (re)training any of the underlying models in our work. Consequently, we cannot run repeated experiments with a given method for a given image, for example by varying a random seed in training or prediction, to obtain a sample size per image. To the best of our knowledge, this would be required for any common statistical test for determining significance on a per image or per dataset basis. Instead, we performed a rank-based analysis over datasets. To the best of our knowledge this is the most suited statistical analysis for our given experimental set-up and is commonly used to compare method performance over multiple datasets (e.g. https://arxiv.org/abs/2305.08252). We have clarified this fact in the revised version of the manuscript.
>
> > Figures are super dense and difficult to interpret.
>
> We have updated the figures and captions to improve readability. In particular, we:
> - Added Figures 3 and 5-8 (Appendix) with detailed qualitative results and Tables 3-7 with all quantitative results.
> - Increased all font sizes and used vector graphics throughout.
> - Improved coherence in Figure 1 a) and extended the respective caption.
> - Simplified and improved the layout of Figure 1 c).
>
> > APG methodology (Section 3.2) is very hard to follow and insufficiently motivated, this would lead to limited novelty relative to prior work.
>
> We have updated this section to provide a better motivation and intuition behind the APG method.
>
> > Lack of qualitative results and visual validation, i.e., ambiguity among cell, nucleus and organoid segmentation claims.
>
> We have added figures with qualitative results, see above.
>
> > Section 3.2 would benefit from an additional conceptual diagram or a simplified explanation of APG that emphasizes intuition over procedural detail.
>
> Figure 1 a) already shows APG conceptually. We have improved this figure and now refer to it earlier in Section 3.2.
>
> > Key parameters should be more clearly defined, including whether they are fixed globally or tuned per dataset, and how sensitive performance is to these choices.
>
> We use the same parameters for all datasets; they are not tuned on a per dataset basis. This was already stated clearly in the paper: “Here, we set $t_{fg}$ = 0.5; $t_{b}$, $t_{c}$, s, and $t_{nms}$ are parameters with default values 0.5, 0.5, 25, and 0.9. We run all experiments with these defaults.” (From Section 3.2). We have slightly rephrased this section for consistency but otherwise don’t see any way to further clarify this.
>
> > Adding qualitative examples across different modalities (e.g., cells vs. nuclei, fluorescence vs. label-free imaging) would substantially strengthen the presentation.
>
> Added figures with qualitative results, see above.
>
> > Some performance claims rely primarily on relative rankings rather than absolute performance differences, which may obscure small but practically meaningful gaps.
>
> Rank-based analysis is appropriate for our experimental setup, see above.

---

### Official Review · Reviewer_Raih · 2026-01-13

**Confidence:** 3
**Preliminary Rating:** 4
**Final Rating:** 4

**Summary:**

The authors bring an always welcome update on the state of the art of cell instance segmentation, by performing a significant number of experiments evaluating a well-defined research question: with the advance of SAM, are specific methods still needed? And answering it by providing a specialized SAM derivative that appears to surpass the general model performance.

**Strengths:**

The paper presents an excellent review of Vision foundation models (VFMs) and their application for cell instance segmentation, and arrives at the simple but powerful idea of combining the ideas from SOA methods, namely combining automatization of SAM with continued fine-tuning and use of prompt based outputs at the same time. There is a genuine algorithmic contribution in leveraging SAM's extensive pre-training for promptable segmentation in the re-run of the segmentation decoding.

The paper is well written in general, with good plots and figures.

**Weaknesses:**

Although experiments are extensive, I missed statistical testing to really clarify in how many of these comparisons the proposed architecture is significantly superior to baseline SAMs and derivatives. Its my impression that it seems to be comparable with current cell instance segmentation specific methods, which is not an impediment for publication but could be made clearer with statistical evidence. Other limitations are already mentioned by the authors.

**Detailed Comments:**

I have no further detailed comments.

**Justification Of Final Rating:**

Appreciate the authors' improvements and clarifications.  Results grounded on detailed statistical analysis, as mentioned in my comments, are necessary to improve my rating, however the authors justified why they are not present. My rating remains the same. I recommend acceptance.

**Justification Of The Preliminary Rating:**

This is amazing work. Missing statistical testing for significance in being better or worse than competing methods makes how APG positions itself against other SOA difficult to understand, and the claim that APG achieves SOA results weaker.

**Questions To Address In The Rebuttal:**

Besides what I mentioned in the weakness, Its confusing to me that the mentioned GitHub in reproducibility (computational-cell-analytics/micro-sam) is already published. But since the method appears derived from micro-sam, that makes sense I suppose. How is micro-sam really different than APG, exactly? I feel like this has to be clearer, or I have missed it.

What is the reasoning in using Mean Segmentation Accuracy with multiple thresholds? This might be something specific to the cell instance segmentation area that I am not familiar with.

---

> ### Author Response · Authors · 2026-01-24
>
> Thank you very much for your careful review and the positive appraisal of our work. We have revised our manuscript based on your and the other reviewer’s feedback.
>
> Regarding the specific points you brought up:
>
> > Although experiments are extensive, I missed statistical testing to really clarify in how many of these comparisons the proposed architecture is significantly superior to baseline SAMs and derivatives.
>
> Please note that we are comparing deterministic instance segmentation methods without (re)training any of the underlying models in our work. Consequently, we cannot run repeated experiments with a given method for a given image, for example by varying a random seed in training or prediction, to obtain a sample size per image. To the best of our knowledge, this would be required for any common statistical test for determining significance on a per image or per dataset basis. Instead, we performed a rank-based analysis over datasets. To the best of our knowledge this is the most suited statistical analysis for our given experimental set-up and is commonly used to compare method performance over multiple datasets (e.g. https://arxiv.org/abs/2305.08252). We have clarified this fact in the revised version of the manuscript.
>
> > Its my impression that it seems to be comparable with current cell instance segmentation specific methods, which is not an impediment for publication but could be made clearer with statistical evidence.
>
> We would summarize the results as follows: APG (our new method) consistently improves over AIS, which are both implemented on top of the same model (μSAM). In some cases this improvement is substantial (e.g. Figure 2 a: TOIAM). APG + μSAM is on par with the state-of-the-art method for cell instance segmentation, CellPoseSAM. CellPoseSAM and APG + μSAM are substantially better than the other models, including CellPose3 and CellSAM, which are specialized models for microscopy segmentation. We have updated statements in the abstract, introduction, and discussion to better reflect this fact.
>
> > It's confusing to me that the mentioned GitHub in reproducibility (computational-cell-analytics/micro-sam) is already published. But since the method appears derived from micro-sam, that makes sense I suppose. How is micro-sam really different than APG, exactly? I feel like this has to be clearer, or I have missed it.
>
> We have implemented APG within the μSAM repository. APG is a different approach to instance segmentation that uses the μSAM model. Specifically, it is different from the original instance segmentation method of μSAM called AIS. AIS is a watershed-based approach (based on predictions from μSAM’s segmentation decoder), whereas APG derives prompts from the segmentation decoder’s predictions and uses these prompts for instance segmentation. So in essence, APG is a (new) part of μSAM that improves its instance segmentation functionality. We have updated the method section to better reflect this fact and have added a new section to μSAM’s documentation: https://computational-cell-analytics.github.io/micro-sam/micro_sam.html#apg (also linked in the paper) to explain how to use APG within μSAM.
>
> > What is the reasoning in using Mean Segmentation Accuracy with multiple thresholds? This might be something specific to the cell instance segmentation area that I am not familiar with.
>
> The Mean Segmentation Accuracy is a standard metric in microscopy instance segmentation. See https://www.nature.com/articles/s41592-023-01942-8 for a detailed discussion of this metric. We now also reference this review in the manuscript.

---

> > ### Comment · Reviewer_Raih · 2026-01-26
> >
> > I appreciate the improvement efforts. Regarding statistical testing, I was mentioning a different type of statistical validation. The 'sample' for a significance test would not be 'multiple runs of the same image', but rather the analysis of the distribution of performance across the $N$ images in the test set. Since the methods are evaluated on the same set of test images, a paired Wilcoxon signed-rank test (after checking for non-normality) or a paired t-test (normal distribution) can be applied image-wise for each dataset. This would provide a p-value to support the claim that Method A is significantly better than Method B on a specific modality or dataset. This is not possible on classification problems but is viable when you have a distribution of paired segmentation results (Image 1: Method A: 0.94, Method B: 0.92).
> >
> > Obviously this would not be a feasible task for one week if you don't have image-wise metrics stored,i.e, you only logged statistics per dataset and not result per image. But if metrics per test image are available, this would strengthen your manuscript's claims.

---

> > > ### Author Response · Authors · 2026-01-27
> > >
> > > Thank you for this comment. We did not consider a statistical analysis based on the distribution of performance across the test set. This kind of analysis is not as common in microscopy segmentation, where statistical performance analysis is typically performed based on variance derived from the model (e.g. due to randomness in training or inference procedure, which typically derives from an i.i.d. distribution) rather than variance across the dataset (which may not be i.i.d. due to the fact that data within a given dataset may come from different experimental conditions etc.). However, we do agree that the statistical analysis you outline is applicable. Unfortunately, we did not store the image level metrics required for this analysis. It is straight-forward to re-predict these results, but due to the large amounts of images we run experiments on this may take some time. We will try to perform this experiment within the discussion phase, but are not yet certain that this can be achieved.

---

> > > > ### Author Response · Authors · 2026-02-01
> > > >
> > > > We managed to implement the statistical evaluation you suggested by applying a ranked Wilcoxon test to the per-image difference in mSA score of two methods. We also checked, the difference is not gaussian in most cases, according to a Shapiro test, so Wilcoxon is more appropriate than a t-test. This enables us to evaluate significance of results per dataset (0.05 significance criterion). We further analyze this by comparing the "wins" (significantly better on a given dataset), "losses" (significantly worse) and "draws" (no significant difference) over all pairs of methods, for the four different modalities.
> > > > You can find the results here: https://owncloud.gwdg.de/index.php/s/MEhXUI1kELy6oq3.
> > > >
> > > > In general, we see that large differences in the scores we reported are always significant, e.g. APG is overall significant better than AMG, CellPose-3, CellSAM, and SAM3, depending somewhat on the domain. AIS, APG, and CellPose-SAM are "closer packed", with the order of evaluation roughly agreeing with our rank-based analysis.
> > > >
> > > > We will add these results to the appendix in the camera-ready version.

---

### Author Rebuttal · Authors · 2026-01-24

**Rebuttal:**

Dear Reviewers, dear Area Chair,

We thank you all for your thorough reviews and the positive appraisal of our work. Based on the reviewer comments, we have made the following changes:

- We have now added qualitative results for all datasets.
- We have also added quantitative results in appendix as a table for all experiments.
- We have made several improvements in figures and manuscript presentation.
- We have carefully addressed feedback from all the reviewers (please see comments below for details).

All changes in the text, including the captions of figures and tables we have added, are marked in red.

**Supporting Material:**

/attachment/63e5c6eb50e7dc20de56a2d36c81811c725e4189.zip

---

### Comment · Area_Chair_BTQP · 2026-01-28

Dear reviewers,

The authors have now responded to your reviews. At this time please participate in discussions with the authors.

IMPORTANT: You must enter your final rating by clicking “Edit” → “Official Review” and providing the Final Rating by February 1st 2026 (23:59 AoE).

Thank you again for your service to MIDL 2026 and making it a success.

---

### Comment · Area_Chair_BTQP · 2026-02-01
**final ratings**

Dear reviewers, if you have not done so already please provide your final ratings for the paper today. If you already did, please disregard this message.
Thanks!

---

### Meta-Review · Area_Chair_BTQP · 2026-02-05

**Recommendation:** Accept (Oral)
**Confidence:** 4

**Metareview:**

The reviewers are all in agreement after the rebuttal of a weak accept rating. There is consensus on the algorithmic novelty (Automatic Prompt Generation method) and practical use for researcher via comprehensive benchmarking. This paper should be of interest to the MIDL community.

---

### Decision · Program_Chairs · 2026-02-13

Accept (Poster)